# The Effects of Circadian Rhythm Disruption on Mental Health and Physiological Responses among Shift Workers and General Population

**DOI:** 10.3390/ijerph17197156

**Published:** 2020-09-30

**Authors:** Mahadir Ahmad, Nur Syafiqa Balqis Md. Din, Ruthra Devi Tharumalay, Normah Che Din, Norhayati Ibrahim, Noh Amit, Nor MF Farah, Rose Azzlinda Osman, Mohamed Faisal Abdul Hamid, Ibnor Azli Ibrahim, Ezad Azraai Jamsari, Mohd Rizal Palil, Sha’ari Ahmad

**Affiliations:** 1Program of Clinical Psychology & Behavioral Health, Faculty of Health Sciences, Universiti Kebangsaan Malaysia, Kuala Lumpur 50300, Malaysia; mahadir@ukm.edu.my (M.A.); nursyafiqabalqis19@gmail.com (N.S.B.M.D.); P92045@siswa.ukm.edu.my (R.D.T.); normahcd@ukm.edu.my (N.C.D.); yatieibra@ukm.edu.my (N.I.); nohamit@ukm.edu.my (N.A.); 2Research Center for Community Health, Faculty of Health Sciences, Universiti Kebangsaan Malaysia, Kuala Lumpur 50300, Malaysia; norfarah@ukm.edu.my; 3Respiratory Unit, Universiti Kebangsaan Malaysia Medical Centre (UKMMC), Jalan Yaacob Latif, Bandar Tun Razak, Kuala Lumpur 56000, Malaysia; roseukm@hotmail.com (R.A.O.); faisal.hamid@ppukm.ukm.edu.my (M.F.A.H.); 4Faculty of Sharia and Law, Universiti Islam Sultan Sharif Ali, Simpang 347, Jalan Pasar Gadong BE 1310, Brunei; azli.ibrahim@unissa.edu.bn; 5Research Centre for Arabic Language and Islamic Civilization, Faculty of Islamic Studies, Universiti Kebangsaan Malaysia, Bangi 43600, Malaysia; ezad@ukm.edu.my; 6Faculty of Economics and Management, Universiti Kebangsaan Malaysia, Bangi 43600, Malaysia; mr_palil@ukm.edu.my

**Keywords:** circadian rhythm, mood, psychological wellbeing, working memory, processing speed, cortisol, triglycerides, glucose

## Abstract

*Background*: The effect of circadian disruption on the bio-psychological clock system has been widely studied. However, the mechanism and the association of circadian rhythm disruption with mental health and physiological responses are still unclear. Therefore, this study was conducted to investigate the effects of circadian rhythm disruption on mental health and physiological responses among shift workers and the general population. *Methods*: A total of 42 subjects participated in this quasi-experimental study. Participants were divided into a group of shift workers (*n* = 20) and a general population group (*n* = 22). Polysomnography tests, blood tests (cortisol, triglycerides and glucose), and psychological tests (Abbreviated Profile of Mood States, General Health Questionnaire-28, Working Memory and Processing Speed Indexes of the Wechsler Adult Intelligent Scale (WAIS-IV) were used to examine the effects of circadian rhythm disruption. *Results*: The results showed a significant relationship between circadian rhythm disruption and mood (*r* = 0.305, *p* < 0.05). The findings of this study also indicated that there was a significant effect of circadian rhythm disruption on mood (F(2,40) = 8.89, *p* < 0.001, η^2^ =0.182), processing speed (F(2,40) = 9.17, *p* < 0.001, η^2^ = 0.186) and working memory (F(2,40) = 4.963, *p* < 0.01, η^2^ = 0.11) in shift workers and the general population. *Conclusions*: Our findings showed that circadian rhythm disruption affects mood and cognitive performance, but it does not significantly affect psychological wellbeing and physiological responses. Future studies are warranted to examine moderator and mediator variables that could influence the circadian rhythm disruption.

## 1. Introduction

Circadian rhythm (CR) is a physiological process with a period of around 24 h, which is essential to regulate the sleep-wake cycle, hormonal secretion, mealtime, body digestion, body temperature, and other significant body functions [1,2]. It also affects how we normally respond to light and dark exposure. This phenomenon occurs through light signaling on the suprachiasmatic nucleus (SCN) in the hypothalamus through the retinohypothalamic tract (RHT) [3]. The hypothalamus responses to the stimulus by secreting arousal hormones such as cortisol. Dark exposure stimulates the eyes to signal the SCN to release melatonin hormone and increase the sleep pressure in the human body. Desynchronization in circadian rhythm occurs when sleep and wake timing is disrupted. The CR disruption is associated with several factors, such as endogenous response, body temperature, and psychological states [3,4]. These factors are positively correlated to the desynchronization of circadian rhythm which will increase the risk of mental and physical health problems [1,3]. Hence, Salgado et al. [5] indicated that there are two crucial factors that contribute to CR disruption, which are the increase of night-time exposure to artificial light and the increase in nocturnal activity that leads to the decrease of sleep time.

CR plays a vital role in balancing mental health, behavioral, and biological changes in humans. Previous research found that CR is associated with hormonal secretion, and it is indirectly linked to mood states, psychological wellbeing, cognitive performance, and physiological changes [6,7,8]. These factors are inter-correlated, where changes in one of the factors might affect other factors associated with CR. Therefore, the role of CR in balancing all factors is one of the keys to healthy well-being. Previous studies indicated that the circadian system regulates the sleep timing of individuals, and sleep is a complex biological process in humans [9]. A complete sleep parameter consists of four stages, which are stage N1, stage N2, stage N3, and stage R sleep [10]. Stage N1 is the first stage of sleep, where the individual presents a low amplitude mixed frequency (LAMF) activity in electroencephalograms (EEG). There is a regular rate of muscle tone and breathing. At this stage, the individual starts to fall asleep. Then, the individual enters stage N2 where the body temperature starts to drop, the heart rate slows down and the sleep spindles and K-complex waves alternate with each other. At this stage, the brain does not perform any mental processes. Stage N3 is the deepest sleep stage where the individual presents a high amplitude signal with a slower frequency. The individual is expected to be in a deep sleep and be difficult to awaken. Then, the individual enters the last stage which is R stage (Rapid Eye Movement stage). During this stage, the individual experiences dreams and is easy to awake. The eyes move faster and brain activity also increases compared to the previous stages.

The desynchronization of CR influences sleep quality and quantity. Therefore, the CR disruption contributes to mental health problems and physiological changes. 

Recent studies have shown that CR disruption is related to many psychological and health-related issues. For instance, the disruption of CR is proven to correlate with psychiatric illnesses such as schizophrenia, depression, neuropsychiatric disorders, and mania [11]. CR disruption also contributes to mood disorders [12]. In addition, different chronic health conditions such as sleep disorders, obesity, diabetes, cardiovascular disease, and seasonal affective disorder are associated with irregular circadian rhythms [13,14,15]. 

Shift working tends to contribute to CR disruption. The hypothesis is based on studies that demonstrate that the shift work environment is often linked to CR misalignment. Shift work is one of the significant contributors to health problems such as cancer, psychiatric disorders, metabolic disorders, and sleep disorders [16]. Evidence from previous studies also showed that the chronic disruption of CR among airline and medical shift workers caused significant physical and mental health problems. It led to mood disturbances and cognitive impairment [11,17]. 

Even though many studies have been conducted to investigate the effects of CR on a specific target population, the effects of CR disruption on mental health and physiological responses are still not well understood. Moreover, previous findings showed inconsistent results with regards to physiological responses such as cortisol, triglycerides, and glucose [18]. Therefore, this study focused on the combination of mental health and physiological markers in assessing the effects of CR disruption on shift workers and the general population. The outcome of this study could contribute to the exploration of psychophysiology and health psychology. It could also help improve shift working systems to minimize the physiological and psychological impacts of exposure to sleep-wake cycle disruption. 

## 2. Materials and Methods 

### 2.1. Study Design

This study employed a quasi-experimental design that divided the research participants into two groups: (i) shift workers and (ii) the general population. In each group, a Pre and Post-design Method was used to compare physiological responses, psychological states, and cognitive performance (mood, psychological wellbeing, working memory, and processing speed) in two conditions: (i) normal sleep and (ii) disrupted sleep conditions. 

### 2.2. Participants 

A total of 42 healthy participants were involved in this study, and they were divided into the group of shift workers (*n* = 20) and the general population group (*n* = 22). The participants were recruited based on the inclusion criteria, which were: healthy adults aged between 20 to 55 years old, literate, with body mass index (BMI) ranging from 18 to 27 kg/m^2^ (BMI is based on the guidelines from the Ministry of Health, Malaysia). The exclusion criteria of this study were: insomnia disorders that cause sleep disturbances, chronic diseases such as diabetes, obesity, hypertension, hyperlipidemia, hypoglycemia, cardiovascular diseases, and mental health problems. The sample size was calculated by using G-Power software. The statistical test power calculation was at the 0.80 level (effect size = 0.5) at the significance level of 0.05 (confidence level = 95%). The minimum sample fixed by this calculation was 12 participants for each group. Written informed consent was provided by the participants prior to their participation in this study. The local ethics committee approved the research for participant protection, following the ethical rules of the Helsinki Declaration 1964 [19]. The study was approved by the Universiti Kebangsaan Malaysia Medical Centre (UKMMC) Ethical Board Committee (JEP-2018-409).

The participants’ characteristics as shown in Table 1 indicated that 77.3% (*n* = 17) were female and 22.7% (*n* = 5) were male in the general population group. In the shift worker group, 75% (*n* = 15) of participants were female and 25% (*n* = 5) were male. In both groups more than half of the participants were married. The mean age of the general population group was 32.82 ± 5.85 years, and 34.40 ± 7.10 years for the shift worker group. The mean BMI of the general population group was 22.75 ± 2.34 kg/m^2^ and the mean BMI of the shift worker group was 22.99 ± 2.82. All participants had work experience of more than one year. The mean duration of the general population group’s work experience was 9.45 ± 6.05 years, and the mean duration of the shift worker group’s work experience was 8.30 ± 5.43 years.

### 2.3. Measures

#### 2.3.1. Polysomnography

Polysomnography (PSG) is a diagnostic tool that measures sleep parameters. It encompasses three vital tools to determine sleep stages, which are electroencephalograms (EEG), electrooculography (EOG), and electromyograms (EMG). This technique is widely used for the evaluation of CR disruptions, which commonly coexist with and exacerbate sleep disorders. Sleep quality was measured by calculating the percentage of deep sleep and light sleep. The diagnosis score of sleep quality is when the deep sleep percentage is more than the light sleep percentage. When the participants’ light sleep percentage was more than their deep sleep percentage, participants were considered to be having sleep disturbance and disrupted CR [10]. This sleep study was conducted at night in a sleep laboratory located at the UKMMC.

#### 2.3.2. Abbreviated Profile of Mood States

The Abbreviated Profile of Mood States is a questionnaire to measure mood disturbances among adults [20]. This questionnaire is made up of 40 items with seven subscales, namely tension-anxiety, depression-dejection, anger-hostility, fatigue-inertia, vigor-activity, confusion-bewilderment, and esteem-related affect. The items are scored by using a Likert scale from 0 (not at all) to 4 (extremely). The Total Mood Disturbance (TMD) was calculated by adding all the negative subscales and subtracting the positive subscale. Then, the score was added with a constant 100 to eliminate the negative value of TMD. A higher score indicates a higher mood disturbance. The reliability of the Malay version showed an excellent internal consistency with Cronbach’s alpha coefficient value of 0.921. 

#### 2.3.3. General Health Questionnaire-28 (GHQ-28)

GHQ-28 is a self-administered screening tool to identify psychological well-being in the general population and within non-psychiatric clinical settings such as primary care or in general medical out-patients [21]. GHQ-28 has four subscales: somatic symptoms, anxiety and insomnia, social dysfunction, and severe depression [22]. Items consist of questions about whether the participant has recently experienced an item behavior or particular symptom rated on a four-point scale. The 28-item version is more commonly used than the other versions (12, 30, 60 items) due to time considerations (less than 5 min), and it has been extensively used in other working populations, allowing for more valid comparisons [23]. GHQ-28 is also reported to have sufficient convergent validity [24], and the reliability of the GHQ Malay Version is reported to have high Cronbach’s alpha (0.91).

#### 2.3.4. Working Memory and Processing Speed Indexes of the Wechsler Adult Intelligent Scale (WAIS-IV)

The WAIS-IV test was developed to measure the cognitive ability of older adolescents and adults. It includes 15 subtests (10 core and five supplemental), which were classified into four major indexes (Verbal Comprehension [VCI], Perceptual Reasoning [PRI], Working Memory [WMI], and Processing Speed [PSI]) [25]. The general intellectual functioning of an individual is measured through the Full-Scale IQ Score (FSIQ). For this study, only the WMI index and PSI index (which comprises of Digit Span, Arithmetic, Symbol Search, and Coding subtests) were administered. 

#### 2.3.5. Blood Test 

A trained phlebotomist performed the cannulation procedure at 8.00 am after the sleep experiment. About five ml of fasting blood was drawn into heparinized and sodium fluoride tubes. The tubes were placed on ice before undergoing centrifugation for plasma separation. An aliquot of plasma was sent to the biochemistry lab for the determination of glucose, triglycerides, and cortisol. These parameters are to indicate physiological responses following a CR disruption. 

### 2.4. Experimental Procedure

This study involved two exposures, which all participants were required to undergo: (i) normal sleep condition (sleep for 8 h) and (ii) disrupted sleep condition (general population group: 5 h of sleep, shift worker group: night shift). In each of these conditions, the study was organized in two parts. The first part required participants to report to the sleep laboratory and spend a night in the laboratory while having their vital functions and sleep activities monitored. The second part involved a blood test and psychological assessments on the following morning when the participant woke up from night sleep. All the psychological tests (APOMS-M, GHQ-28, WMI (WAIS-IV) and PSI (WAIS-IV) were administered at two different time points, namely before and after circadian rhythm disruption. The study flow chart is indicated in Figure 1. 

#### 2.4.1. Normal Condition

For the normal condition, participants arrived at the sleep laboratory in the evening between 08.00–09.00 p.m. The sleep laboratory at the UKMMC is equipped with a sleep monitoring system (Somnotec, Selangor, Malaysia) and a video telemetry device (Somnotec, Selangor, Malaysia) to monitor participants’ sleep throughout the night. The sleep laboratory consists of individual rooms that mimic a home environment. Each sleep test session involved polysomnography measurements in which respiratory variables (e.g., effort to breathe, airflow, and oxygen saturation), cardiac variables (e.g., heart rate), and EEG were measured to determine the responses during sleep, in the sleep laboratory. Participants were asked to lie on the bed by 10:00 p.m. Once the participants were ready, a polysomnography device was attached to the body. Accurate measurement of the skull was measured based on the 10–20 system. EEG electrodes (Somnotec, Selangor, Malaysia) were attached to the scalp region of the cortex between the motor and sensory cortex (C3 and C4) and at the occipital lobe (O1 and O2). Meanwhile, the EOG electrodes were placed at the left and right outer eye with one above and one beneath the horizontal plane. Three EMG electrodes were placed beneath the chin overlying the mentalis and submental muscles. These muscles are particularly sensitive to changes in muscle tone. All personal electronic gadgets such as mobile phone were turned off when in bed. The room light was switched off to indicate the time to sleep. 

In the normal sleep condition exposure, participants were required to sleep for 8 h, and following that, they were woken up at 6.00 a.m. by the sound of an alarm. Once they were awake, the polysomnography devices were detached. Participants were given an hour to prepare themselves for the second part of the experiment, which was the blood test and psychological assessments. Then, the participants underwent a series of psychological and cognitive assessments to evaluate their psychological health and cognitive performance. The participants underwent four simple tests which took about 45 min to complete: (1) Working Memory Test—the participants were tested with the Digit Span and Arithmetic subtests of the WAIS-IV. (2) Processing Speed Test—the participants were tested with the Symbol Search and Coding subtests of the WAIS-IV. (3) Mood State Test—the mood of the participants was measured using the Abbreviated Profile of Mood States Malay version (APOMS-M). (4) Psychological Wellbeing Test—the level of psychological well-being of the participants was assessed using the General Health Questionnaire-28 (GHQ-28). Every group underwent the same procedures. 

#### 2.4.2. Disrupted Condition

In this condition, participants from the two different groups (general population/shift workers) underwent slightly different conditions to simulate a disrupted sleep condition. The procedures for each group are described separately as follows:

#### 2.4.3. General Population

Participants reported to the sleep laboratory between 10.00–11.00 p.m. They were fitted with the polysomnography devices as described previously. They were required to sleep at 1.00 a.m. and be woken up at 6.00 a.m. by the sound of an alarm. The total amount of sleep was 5 h. Participants underwent the same blood test and psychological assessments that were described previously.

#### 2.4.4. Shift Worker

Participants were required to report to the sleep laboratory within one hour after completing their night shift. Since the night shift typically ends at 7.00 a.m. therefore, the test began at 8.00 a.m. No polysomnography test was required as no sleep was induced. Once the first test was completed, the participants immediately underwent the same blood tests and psychological assessments that were described previously.

### 2.5. Statistical Analysis

To identify the relationship of sleep quality with mental health (mood, psychological wellbeing, working memory, processing speed) and physiological responses (glucose, triglycerides, cortisol), Kendal Tau b analysis was used to identify the relationship between variables. To determine the effect of CR on mood, psychological wellbeing, working memory, processing speed, glucose, triglycerides, and cortisol in the shift worker group and the general population group, a mixed-design ANOVA analysis was employed. The data were analyzed by using the IBM SPSS version 25 software (IBM Corp., Armonk, NY, USA).

## 3. Result

Table 2 shows the general profile of mental health and cognitive performance for both groups. The mean mood of the general population group post-test (92.59 ± 12.69) was slightly higher than pre-test (88.91 ± 17.87). The mean mood of the shift worker group also showed a higher score in post-test (93.20 ± 23.30) as compared to pre-test (83.45 ± 21.56). All participants’ mental health scores were normal before and after circadian rhythm disruption; the total GHQ-28 scores were below the score of 23 and classified as non-psychiatric. The working memory index (WMI) and processing speed index (PSI) of participants for the pre-post test showed that all participants were at average level. The effects of circadian rhythm disruption on mental health and cognitive performance will be explained in detail in the next section. 

Table 3 shows the mean and standard deviation for all participants’ blood variables. The mean of morning serum cortisol level for post-test (261.78 ± 82.80) in the general population group was higher than pre-test (240.85 ± 64.24). In the shift group, the cortisol level for post-test (242.07 ± 126.10) was lower than pre-test (321.36 ± 103.27). Both groups showed a normal range of cortisol serum level in the pre-post test. The mean of triglycerides for pre-test (0.98 ± 0.67) and post-test (1.03 ± 0.60) in the general population group was within the normal range. The shift worker group also showed normal triglycerides levels for pre-test (1.12 ± 0.53) and post-test (1.02 ± 0.70). The glucose levels among the general population group were also in normal range for pre-test (5.02 ± 0.42) and post-test (4.96 ± 0.43). The shift worker group also showed a normal range of glucose levels at pre-test (4.74 ± 0.68) and post-test (4.78 ± 0.61) The effects of circadian rhythm disruption on physiological responses will be elaborated in the next section. 

Table 4 shows the sleep quality percentage among the general population group. The results indicated that almost all participants (90.1%) achieved a good sleep quality after sleep for eight hours. However, after participants were exposed to circadian rhythm disruption, only 40.9% of participants experienced good sleep quality, while more than half of the participants (59.1%) had poor sleep quality. 

### 3.1. The Relationship between Circadian Rhythm and Mental Health (Mood, Psychological Wellbeing, Working Memory, and Processing Speed) 

The result of the Kendall Tau-b analysis showed that there was a significant positive correlation between CR disruption and mood states (*r* = 0.305, *p* < 0.05); a significant negative relationship between glucose and processing speed (*r* = −0.219, *p* < 0.05); and a significant positive correlation between cortisol and mood (*r* = 0.263, *p* < 0.05) (Table 5).

### 3.2. The Effect of Circadian Rhythm Disruption on Mood

A 2 × 2 mixed-design ANOVA was performed to identify the effect of CR disruption on the mood state. All 42 participants from the two different groups were asked to answer the POMS questionnaire before and after being exposed to the CR disruption. A significant main effect of CR disruption on mood was obtained [F(2,40) = 8.89, *p* < 0.001, η^2^ = 0.182]. A significant main effect of group was not found [F(2,40) = 0.197, *p* = 0.659, η^2^ = 0.01]. A non-significant interaction between condition and group was reported [F(2,40) = 1075.9, *p* = 0.186, η^2^ = 0.04]. The means of the two groups indicated that although there was a significant change in the condition of the general population group before disruption (M = 88.91, SD = 17.87) to after the disruption (M = 92.59, SD = 12.69), there were no corresponding changes in the condition of the shift worker group from before the disruption (M = 83.45, SD = 21.56) to after disruption (M = 93.20, SD = 23.30) (Table 6 and Figure 2).

### 3.3. The Effect of Circadian Rhythm Disruption on Psychological Wellbeing

The results showed that there was no significant main effect of CR disruption on the psychological wellbeing of the participants [F(2,40) = 0.005, *p* = 0.942, η^2^ = 0.00]. This result indicated that the profile of psychological wellbeing was not influenced by the disruption of circadian rhythm in both groups. A non-significant main effect of group was reported [F(2,40) = 0.846, *p* = 0.363, η^2^ = 0.021]. A non-significant interaction between condition and group toward psychological wellbeing was reported [F(2, 40) = 1.97, *p* = 0.169, η^2^ = 0.05]. The means of the two groups indicated that there was no interaction in the general population group before the CR disruption (M = 1.05, SD = 1.68) and after the disruption (M = 1.55, SD = 2.24) and also in the shift worker group before the disruption (M = 2.05, SD = 2.54) and after CR disruption (M = 1.60, SD = 2.14) (Table 6 and Figure 3). 

### 3.4. The Effect of Circadian Rhythm Disruption on Cognitive Performance (Processing Speed and Working Memory)

The analysis found that there was a significant main effect of CR disruption on processing speed [F(2,40) = 9.17, *p* < 0.001, η^2^ = 0.186]. The result also showed that there was no significant main effect of processing speed on the group [F(2,40) = 2.003, *p* = 0.165, η^2^ = 0.048]. There was no significant interaction between the condition of CR disruption with group [F(2,40)= 0.000, *p* = 0.984, η^2^ = 0.000]. The means of the two groups indicated that although there was no significant change of processing speed in the condition of the general population group before disruption (M = 101.68, SD = 9.46) to after the disruption (M = 106.85, SD = 13.29) as well as no changes in the condition the of shift worker group within the period of before the disruption (M = 106.70, SD = 11.99) and after the disruption (M = 111.95, SD = 15.98) (Table 6 and Figure 4). The result also showed a significant main effect of CR disruption on working memory [F(2,40) = 4.963, *p* < 0.01, η^2^ = 0.11]; a non-significant main effect of working memory on groups [F(2,40) = 2.061, *p* = 0.159, η^2^ = 0.049]; no significant interaction effect of CR disruption with groups [F(2,40) = 0.078, *p* = 0.782, η^2^ = 0.002]. The means indicated that the condition before exposing to CR disruption (M = 94.09, SD = 11.01) and after exposed to CR disruption (M = 97.5, SD = 15.36) in the general population group did not make any changes as also showed in the shift worker group before being exposed to CR disruption (M = 89.10, SD = 11.90) and after (M = 91.75, SD = 12.75) (Table 6 and Figure 5).

### 3.5. The Effect of Circadian Rhythm Disruption on Physiological Responses (Glucose, Triglycerides, Cortisol)

The result indicated that there was no significant main effect of CR disruption on glucose [F(2,39) = 291, *p* = 0.592, η^2^ = 0.007], triglycerides [F(2,39) = 0.035, *p* = 0.852, η^2^ = 0.001], or cortisol [F(2,23) = 1.89, *p* = 0.182, η^2^ = 0.076]. There was a significant interaction effect of CR disruption on cortisol [F(2,25) = 5.47, *p* = 0.028, η^2^ = 0.192], which indicates that the CR disruption condition and the group could influence the cortisol level. Mean cortisol levels in the general population group (M = 240.85, SD = 64.24) were lower than in the shift workers group (M = 311.33, SD = 108.66) for normal condition. Meanwhile, the mean cortisol levels in the disrupted condition were higher in the general population group (M = 261.54, SD = 81.73) than in the shift worker group (M = 231.25, SD = 135.17) (Figure 6). No interaction effects of CR disruption were noted for triglycerides [F(2,39) = 1.02, *p* = 0.319, η^2^ = 0.025] (Figure 7) and glucose [F(2,39) = 0.032, *p* = 0.86, n^2^ = 0.001] (Table 6 and Figure 8). 

## 4. Discussion

The effects of CR disruption on mental health and physiological responses have been widely studied by researchers over the years [1,7,15]. However, this study focused on the combination of mental health and physiological markers in testing the effects of CR disruption. The significant effect of CR disruption on mood indicated that the one-night total sleep disturbance consistently affects the total mood disturbance of an individual. Our result was in parallel with a study conducted by Bechtel [12], which indicated that there is a positive correlation between mood and CR disruption. According to Saghir et al. [8] the alteration of mood states is related to the function of the amygdala. As the amygdala is the emotional center of the brain, it plays a primary role in changing the mood state of an individual. When individuals experience CR disruption, there is a functional imbalance between the ventral anterior cingulate cortex and the amygdala; the changes in mood may cause the amygdala to increase the response to unhealthy stimuli [8,26]. The result also showed that mood states were sensitive to the changes in the sleep and wake cycle.

This study also supports a study conducted by Kaida and Niki [27], which indicated that sleep disturbance is likely to disturb CR and simultaneously reduce positive mood and increase negative mood. In addition, the result is also consistent with the study conducted by Babson et al. [28], which pointed out that interruption to sleep increases symptoms of anxiety and depression. Meanwhile, Zohar et al. [29] indicated that interruption to sleep enhances negative reactions to life-threatening events while decreasing positive reactions to happy events. Our study also indicated that there was no correlation between CR disruption and psychological wellbeing, cognitive performances (processing speed, working memory), or physiological responses (cortisol, glucose, and triglycerides). This might be due to the endogenous adaptation of the CR disruption among participants [30]. Participants who reacted positively with the strategies to promote adjustment to the CR disruption might not be influenced by their mental health and physiological responses [29,30]. In addition, the consideration of individual chronotype in determining the work rotation schedule could also influence the effect of CR disruption on mental health and physiological responses [31]. Our results also showed that there was a significant difference between the shift worker group and the general population group in mood, working memory, and processing speeds before and after CR disruptions. The result indicated that both groups had higher mood states, working memory, and processing speeds after CR disruption. In the general population group, an increase in mood states among the participants could be due to the normal regulation of the Hypothalamic Pituitary Adrenal (HPA) axis system [32]. It starts from the release of corticotrophin-releasing factor (CRF) from the hypothalamus which binds with the CRF receptors. Then, the released adrenocorticotropic hormone (ACTH) binds with the adrenal cortex receptor and stimulates cortisol hormone production in the adrenal cortex [32,33]. However, this effect was not exhibited in the shift worker group, as the trend showed lower cortisol levels after CR disruption. The reduction of cortisol levels can be associated with the endocrine homeostasis since the endocrine system plays an essential role in hormone regulation and homeostasis. Thus, we showed that the cortisol hormone was influenced by the environmental stimulus, which was stimulated by the CR disruption. Our finding is consistent with the study conducted by Kudielka et al. [34], which demonstrated that low quality of sleep was related to a lower level of cortisol. 

In shift workers who positively adapt to a schedule rotation, it may not influence their mental health status and they may be able to readjust their CR in a short time [35,36]. Meanwhile, the increase in cognitive performance is characterized by the improvement of cognitive control among participants, which influences the increase of working memory and processing speed among participants [29]. The cognitive control of participants may increase the motivation of individuals. When the motivation of participants is increased, it could also influence the increase of cognitive performance [29,36]. Thus, our results showed that there were no significant differences between the general population group and the shift worker group on the effect of CR disruption on mood, psychological wellbeing, working memory, processing speed, cortisol, triglycerides, and glucose. Both groups showed almost the same pattern of the effects of CR disruption. This result is in line with a previous study conducted by Ghiasvand [37], which indicated that there was no correlation between triglycerides and glucose in shift workers and day workers. Our finding is also strengthened with similar results observed in a study conducted by Van Amelsvoort et al. [38], which explain that the forward shift rotation system is a good system in controlling the effect of CR disruption psychologically. The forward shift system and a positive environment impede the negative effect of CR disruption.

We also found that there was no negative effect of CR on psychological wellbeing among the participants. This indicates that sleep disruption for only one night (sleep less than 5 h) did not influence the general mental health status of individuals. However, repetitive sleep disruption (sleep debt) could increase the negative effect of CR disruption [8,38]. Many studies indicated that sleep debt triggers the activation of CR disruption and leads to mental health problems. For instance, a study conducted by Regestein et al. [39] showed that the accumulation of 2 h of sleep deprivation a day for five days could increase melancholic symptoms in individuals. The result on shift workers indicates that the existing forward shift rotation system is a suitable system to be practiced in the study population.

In this study, we did not find any significant effect of CR disruption on cortisol, triglycerides, or glucose. However, the decrease in total cortisol, triglycerides, and glucose response toward CR disruption in shift workers may indicate that shift workers would be exposed to the long-term effects of circadian misalignment. A study by Kim et al. [7] also showed that the long-term effect of shift work is associated with a decrease in cortisol and glucose tolerance response. Fonken et al. [40] reported that dim light provokes the alteration of metabolic response toward CR disruption, including the reduction of glucose tolerance at night. Results also showed that there was an increase of cortisol and triglycerides response toward CR disruption in the normal group. Therefore, this result indicated that 5 h of sleep might increase vulnerability to health problems. Nonetheless, there were two important limitations of this study. Firstly, the small sample size could influence the result of the study. Secondly, this study only implemented the pre-post study design with potential exposure to the influence of the unknown confounding factors. 

## 5. Conclusions

CR disruption shows various impacts on human life. Our study suggests that the main and interaction effects of CR disruption on mental health and physiological response can occur in many ways, which can be perceived as a complex association. However, this study agreed that the mood state was consistently affected by the disruption of CR. Nevertheless, we need a clear explanation of the mediator and moderator variables that influence the circadian rhythm disruption effect. Future research should explore the comorbid features that influence the result of this study in detail. 

## Figures and Tables

**Figure 1 ijerph-17-07156-f001:**
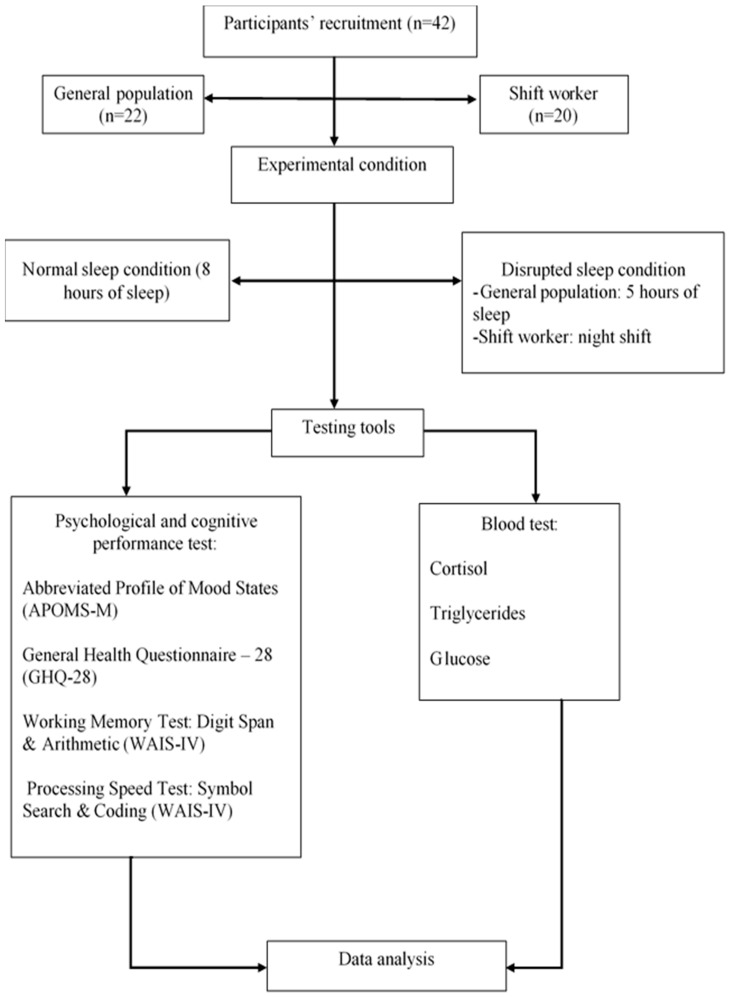
The Study’s Flow Chart.

**Figure 2 ijerph-17-07156-f002:**
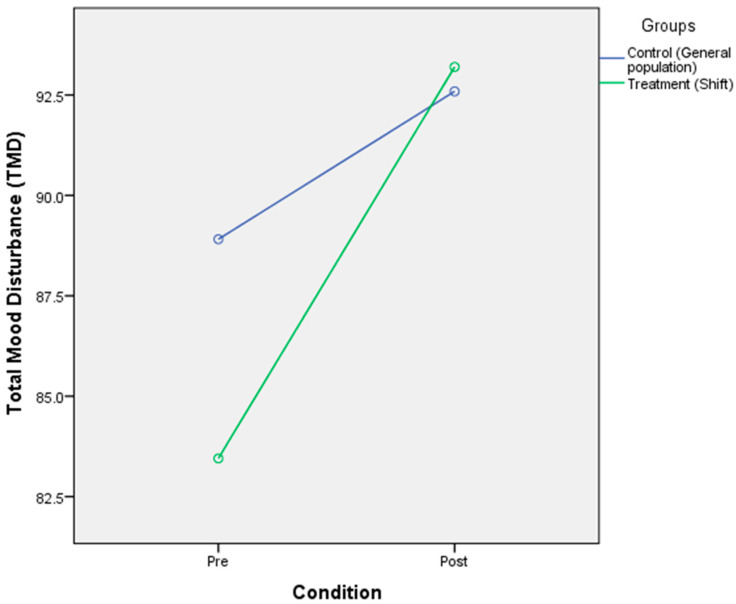
Mixed-design ANOVA analysis of circadian rhythm disruption on the mood states.

**Figure 3 ijerph-17-07156-f003:**
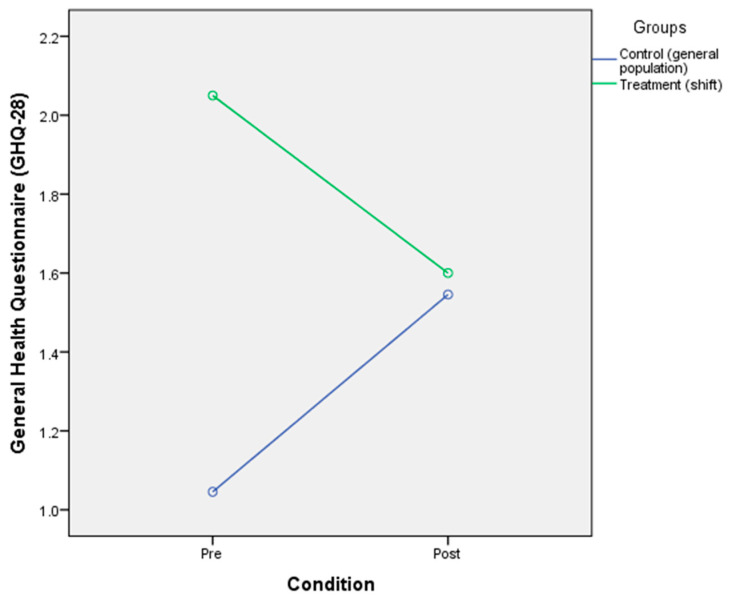
Mixed-design ANOVA analysis of circadian rhythm disruption on psychological wellbeing.

**Figure 4 ijerph-17-07156-f004:**
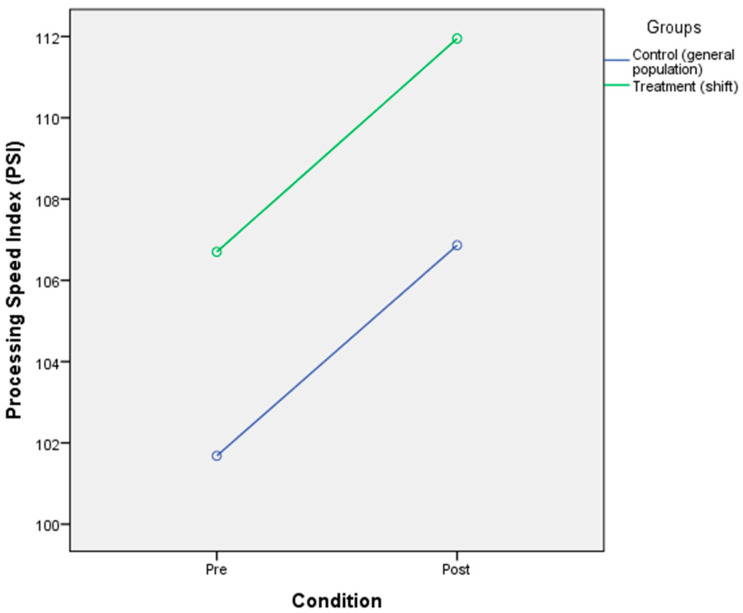
Mixed-design ANOVA analysis of circadian rhythm disruption on processing speed.

**Figure 5 ijerph-17-07156-f005:**
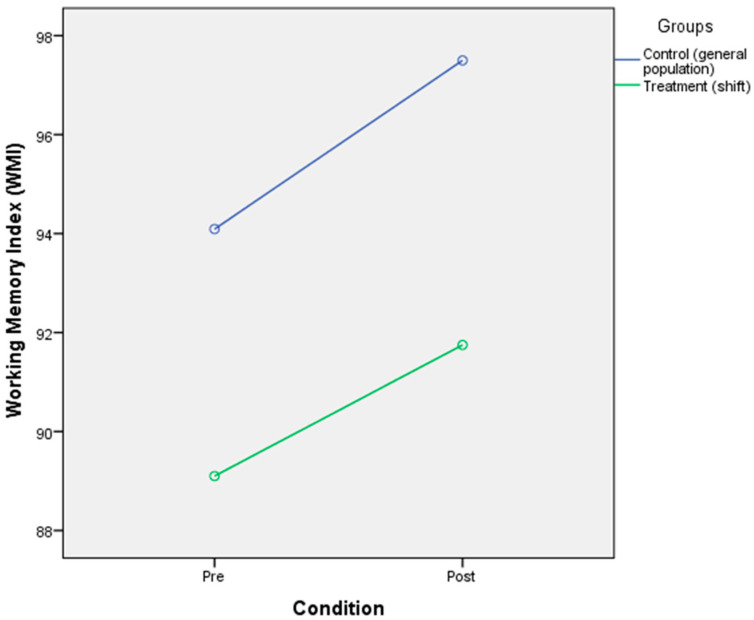
Mixed-design ANOVA analysis of circadian rhythm disruption on the working memory.

**Figure 6 ijerph-17-07156-f006:**
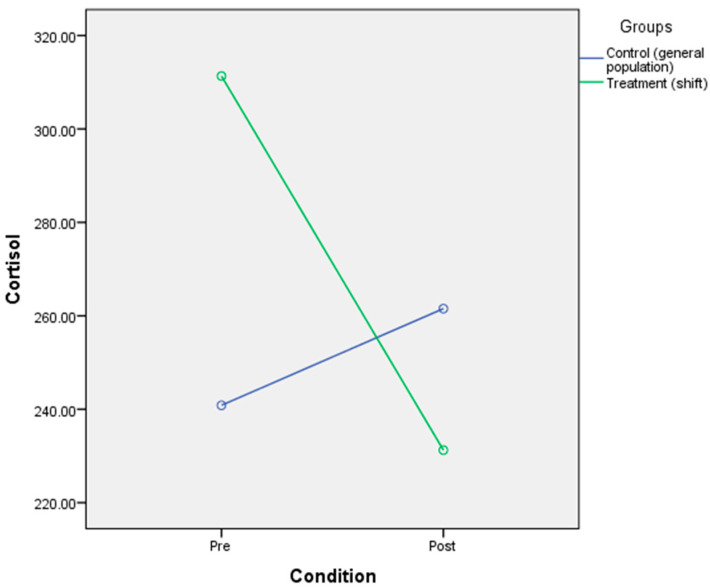
Mixed-design ANOVA analysis of circadian rhythm disruption on cortisol.

**Figure 7 ijerph-17-07156-f007:**
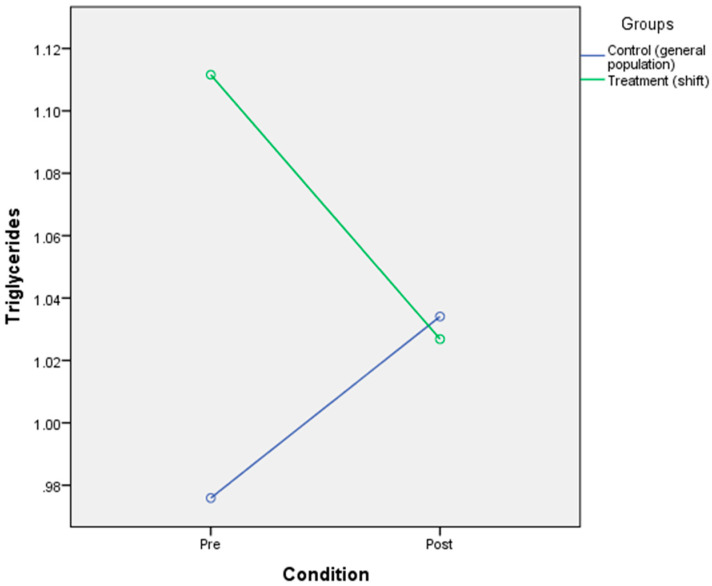
Mixed-design ANOVA analysis of circadian rhythm disruption on triglycerides.

**Figure 8 ijerph-17-07156-f008:**
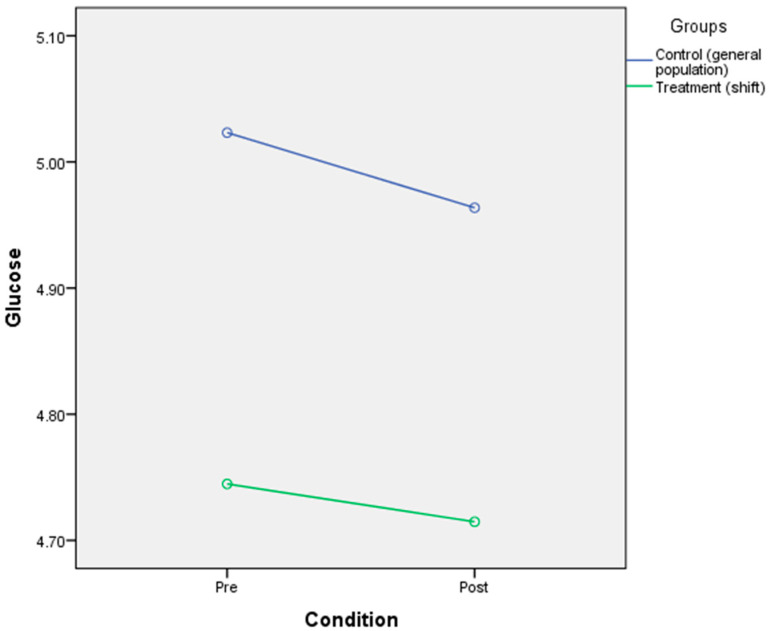
Mixed-design ANOVA analysis of circadian rhythm disruption on glucose.

**Table 1 ijerph-17-07156-t001:** Participants’ characteristics.

Demography	General Population (*n* = 22)		Shift Worker (*n* = 20)	
	Frequency (Percentage)	Mean ± SD	Frequency (Percentage)	Mean ± SD
**Sex**				
Male	5 (22.7%)		5(25%)	
Female	17(77.3%)		15 (75%)	
**Marital status**				
Single	9(41%)		7(35%)	
Married	13(59%)		13(65%)	
**Age**				
20 to 29	7(31.8%)		8(40%)	
30 to 39	11(50%)	32.82 ± 5.85	3(15%)	34.40 ± 7.10
40 to 49	4(18.2%)		9(45%)	
**Body Mass Index (BMI)**				
18.0 to 21.9	9(41%)		8(40%)	
22.0 to 25.9	11(50%)	22.75 ± 2.34	9(45%)	22.99 ± 2.82
26.0 to 29.9	2(9%)		3(15%)	
**Working Experience**				
1 to 5 years	8(36.4%)		8(40%)	
6 to 10 years	6(27.3%)		6(30%)	
11 to 15 years	3(13.6%)	9.45 ± 6.05	3(15%)	8.30 ± 5.43
16 to 20 years	4(18.2%)		3(15%)	
21 to 25 years	1(4.5%)		0	

**Table 2 ijerph-17-07156-t002:** Profiles of mental health and cognitive performance (*n* = 42).

Parameter	General Population		Shift Worker	
	Mean ± SD (Pre)	Mean ± SD (Post)	Mean ± SD (Pre)	Mean ± SD (Post)
Mood	88.91 ± 17.87	92.59 ± 12.69	83.45 ± 21.56	93.20 ± 23.30
Psychological wellbeing	1.05 ± 1.68	1.55 ± 2.24	2.05 ± 2.54	1.60 ± 2.14
Working memory	94.09 ± 97.50	97.50 ± 15.36	89.10 ± 11.90	91.75 ± 12.75
Processing speed	101.68 ± 9.46	106.85 ± 13.29	106.70 ± 11.99	111.95 ± 15.98

**Table 3 ijerph-17-07156-t003:** Participants’ blood variables.

Parameter	General Population		Shift Worker	
	Mean ± SD (Pre)	Mean ± SD (Post)	Mean ± SD (Pre)	Mean ± SD (Post)
Cortisol	240.85 ± 64.24	261.54 ± 81.73	311.33 ± 108.66	231.25 ± 135.17
Triglycerides	0.98 ± 0.67	1.03 ± 0.60	1.11 ± 0.54	1.02 ± 0.70
Glucose	5.02 ± 0.42	4.96 ± 0.43	4.74 ± 0.68	4.71 ± 0.55

Note: All blood variables were measured in nmol/L (SI units).

**Table 4 ijerph-17-07156-t004:** Profile of sleep quality among the general population group (*n* = 22).

Sleep Quality Profile		Pre-Test % (*n*)	Post-Test % (*n*)
Sleep quality	Good sleep	90.1% (20)	40.9% (9)
	Poor sleep	9.9% (2)	59.1% (13)

**Table 5 ijerph-17-07156-t005:** The relationship between circadian rhythm disruption and mood, psychological wellbeing, working memory, processing speed, glucose, triglycerides, and cortisol.

Research Variables		1	2	3	4	5	6	7	8
1	Circadian rhythm disruption								
2	Mood	0.305 *							
3	Psychological wellbeing	0.028	−0.042						
4	Working Memory	−0.114	−0.178	0.026					
5	Processing Speed	−0.167	−0.005	0.026	0.034				
6	Glucose	0.091	−0.183	−0.238	0.049	−0.219 *			
7	Triglycerides	−0.046	−0.098	−0.162	0.118	−0.088	0.212		
8	Cortisol	0.074	0.263 *	−0.183	−0.078	0.097	0.089	−0.116	

* *p* < 0.05 significant value.

**Table 6 ijerph-17-07156-t006:** Summary of the result of mixed-design ANOVA.

Parameter	General Population		Shift Worker		*p* Value	*p* Value
	Mean ± SD (Pre)	Mean ± SD (Post)	Mean ± SD (Pre)	Mean ± SD (Post)	(Within)	(Between)
Mood	88.91 ± 17.87	92.59 ± 12.69	83.45 ± 21.56	93.20 ± 23.30	0.005 **	0.659
Psychological wellbeing	1.05 ± 1.68	1.55 ± 2.24	2.05 ± 2.54	1.60 ± 2.14	0.942	0.363
Working memory	94.09 ± 11.01	97.50 ± 15.36	89.10 ± 11.90	91.75 ± 12.75	0.032 *	0.159
Processing speed	101.68 ± 9.46	106.85 ± 13.29	106.70 ± 11.99	111.95 ± 15.98	0.004 **	0.165
Cortisol	240.85 ± 64.24	261.54 ± 81.73	311.33 ± 108.66	231.25 ± 135.17	0.182	0.557
Triglycerides	0.98 ± 0.67	1.03 ± 0.60	1.11 ± 0.54	1.02 ± 0.70	0.852	0.729
Glucose	5.02 ± 0.42	4.96 ± 0.43	4.74 ± 0.68	4.71 ± 0.55	0.592	0.069

* *p* < 0.05 significant value; ** *p* < 0.01 significant value

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
