# Peer review of "The Effects of Circadian Rhythm Disruption on Mental Health and Physiological Responses among Shift Workers and General Population"

_ijerph, 2020, doi:10.3390/ijerph17197156_

Round 1
Reviewer 1 Report
Reviewer comments and suggestions
The present study by Ahmad et al. investigated the effects of circadian rhythm disruption on mental and physiological responses among shift workers and the general population. The study used various techniques to explore the several parameters of sleep disorders. According to their study, they have utilized data of 42 subjects and polysomnography test, blood test (cortisol, triglycerides, and glucose) and psychological tests (Abbreviated Profile of Mood States, General Health Questionnaire – 28, Working Memory and Processing Speed Indexes of the Wechsler Adult Intelligent Scale (WAIS-IV) were recorded. Finally, the result showed a significant relationship between circadian rhythm disruption and mood (r=.305, 34 p<.05). The findings of this study also indicated that the effect of circadian rhythm disruption on mood (F(2, 40)=8.89, p<0.001, n2 = .182), processing speed(F(2,40)=9.17, p<.001, n2=.186) and working memory(F(2,40)=4.963, p<.01, n2=.11) in shift worker and the general population. This paper highlighted that circadian rhythm disruption affects mood and cognitive performances.
The manuscript needs to modify based on the below-suggested comments before taking the final decision on the manuscript.
- Line 29, it is needed to present in the abstract about how many of them were shift worker and the general population
- Line 35-36, please rewrite the line as it looks grammatical after reading.
- Line 39, there was no point related to physiological response in the result part
- Line 53-54, need an explanation for the line, elaborate
- Line 65, the stage needs to be explain in 2-3 lines
- Line 82-83, Need a suitable reference for the line,
- Line 159, how the persons are subjected to DSC
- Line 217, the numerical representation was wrong, please check
- Table 2, is it possible to include a table that can also be divided into two groups normal and shift worker
- Table 3, there was no unit provided for the biochemical parameters
- Line 267-268, is there was any specific reason for this
- Why the figures are not in order form
- Line 327, what does it mean?
- Please highlight the specific parameter of mood defining? (line 329)
- Line 337, and line 339, the same number of reference by a different author
- Line 343-345, please provide a reason for this
- Line 369-371, why the results were insignificant
- Related to cortisol, 390, one study was in contrary of your finding “impact of shift work on the diurnal cortisol rhythm: a one-year longitudinal study in junior physicians Jian Li, Martin Bidlingmaier,2 Raluca Petru,3 Francisco Pedrosa Gil,4 Adrian Loerbroks,1 and Peter Angerer1” please discuss
- The author should write the limitations of the study
- Please check the references as it was not formatted according to the journal guidelines.
Reviewer 2 Report
The aim of the paper “is to investigate the effects of circadian rhythm disruption on mental health and physiological responses among shift workers and the general population”.
The aims of the research, and the methods (a quasi-experimental study) are not clearly described in the paper. Also the outcome is not clear.
There is confusion between mood, psychological wellbeing, mental health, health psychology.
The references are not up to date, and measures references are not always correct.
The abstract does not follow the criteria indicated by the journal (subdivision into background, methods, results, conclusions).
The experimental procedure is not well described. It may be helpful to include a flow chart.
The results are similarly unclear.
The discussion and the conclusion could be discussed more deeply, after the changes required
Specific comments
Line 124 The correct reference is 19, not 18, but is not recent (1992)
Line 132 GHQ- 28 needs a check ; here it’s used for identify “minor psychiatric disorders”?
Line 141 reference 26 is not correct, the correct should be 23
After line 124 all references must be corrected
BMI ?????
Line 191 GHQ- 28 needs a check ; here it’s used to measure “psychological well being”
Line 231 mean and standard deviation (???) , of blood variables ???
Line 157 Experimental Procedure needs a specific section to permit the readers to understand the design of the research.
Line 216 Results: the characteristics of the participants and the related table should not be placed in the results, but in materials and methods, or sample.
Table 1: all the sample ? there is no data on shift workers and the general population
Line 222 The characteristics of partecipants are not described for subgroups
Line 216 After 3. Results we find 4.1. The relationship between circadian rhythm and mental health (mood, psychological wellbeing, working memory, and processing speed) and 4.2. The effect of circadian rhythm disruption on mood. Where is 4 ?
Line 251
Line 254 POMS ? “The POMS questionnaire before and after being exposed to the CR disruption) are not described in the measures
From line 251 to 279 : why did you do the analyzes on all subjects?
Round 2
Reviewer 2 Report
The aim of the paper “is to investigate the effects of circadian rhythm disruption on mental health and physiological responses among shift workers and the general population”.
The aims of the research, the methods and the outcome are more clear now.
The references are now correct.
The abstract follows the criteria indicated by the journal (subdivision into background, methods, results, conclusions).
The experimental procedure is now well described and the Authors included a flow chart.
Adequate responses were given to specific comments.